# Stereotactic Body Radiotherapy (SBRT) to Localised Prostate Cancer in the Era of MRI-Guided Adaptive Radiotherapy: Doses Delivered in the HERMES Trial Comparing Two- and Five-Fraction Treatments

**DOI:** 10.3390/cancers16112073

**Published:** 2024-05-30

**Authors:** Rosalyne Westley, Francis Casey, Adam Mitchell, Sophie Alexander, Simeon Nill, Julia Murray, Ragu Ratnakumaran, Angela Pathmanathan, Uwe Oelfke, Alex Dunlop, Alison C. Tree

**Affiliations:** 1The Royal Marsden NHS Foundation Trust, London SM2 5PT, UK; 2Radiotherapy and Imaging Division, Institute of Cancer Research, London SM2 5NG, UK; 3Joint Department of Physics, The Institute of Cancer Research and The Royal Marsden NHS Foundation Trust, London SM2 5PT, UK

**Keywords:** SBRT, prostate radiotherapy, MRI-guided adaptive radiotherapy, intrafraction motion

## Abstract

**Simple Summary:**

Radiotherapy to the prostate with curative intent was previously a procedure that was delivered over seven and a half weeks. The standard of care is now 20 treatments over 4 weeks; however, 5 treatments have recently been shown to be as good as 20 treatments in patients with intermediate risk disease. Fewer treatments mean that patients spend less time in the hospital and travelling to appointments and helps to reduce the burden on radiotherapy departments. When reducing the number of treatments, the dose per treatment is increased. This requires greater precision and certainty of dose delivery. The magnetic resonance linear accelerator (MRL) delivers radiotherapy under MRI guidance with the ability to modify the plan to adapt to any changes in the patient’s anatomy, thus potentially enabling the safe delivery of higher doses per treatment. This study addresses the true dose delivered in two and five treatments of the MRL and whether target objects are being met.

**Abstract:**

HERMES is a phase II trial of MRI-guided daily-adaptive radiotherapy (MRIgART) randomising men with localised prostate cancer to either 2-fractions of SBRT with a boost to the tumour or 5-fraction SBRT. In the context of this highly innovative regime the dose delivered must be carefully considered. The first ten patients recruited to HERMES were analysed in order to establish the dose received by the targets and organs at risk (OARS) in the context of intrafraction motion. A regression analysis was performed to measure how the volume of air within the rectum might further impact rectal dose secondary to the electron return effect (ERE). One hundred percent of CTV target objectives were achieved on the MRI taken prior to beam-on-time. The post-delivery MRI showed that high-dose CTV coverage was achieved in 90% of sub-fractions (each fraction is delivered in two sub-fractions) in the 2-fraction cohort and in 88% of fractions the 5-fraction cohort. Rectal D1 cm^3^ was the most exceeded constraint; three patients exceeded the D1 cm^3^ < 20.8 Gy in the 2-fraction cohort and one patient exceeded the D1 cm^3^ < 36 Gy in the 5-fraction cohort. The volume of rectal gas within 1 cm of the prostate was directly proportional to the increase in rectal D1 cm^3^, with a strong (R = 0.69) and very strong (R = 0.90) correlation in the 2-fraction and 5-fraction cohort respectively. Dose delivery specified in HERMES is feasible, although for some patients delivered doses to both target and OARs may vary from those planned.

## 1. Introduction

The low α/β ratio of prostate cancer combined with technical advances in radiotherapy delivery has motivated the adoption of hypofractionated regimes, with the aim of achieving a higher effective dose to the prostate without increasing acute and late toxicities [1,2,3].

Although 60 Gray (Gy) in 20 fractions is considered standard of care [4,5,6] level 1 evidence now supports stereotactic body radiation therapy (SBRT) in five fractions with biochemical non-inferiority for early-stage disease and favourable toxicity outcomes [7,8,9,10]. The next logical step is to examine whether radical treatment can be delivered in less than five fractions, making accurate dose delivery ever more critical. The delivery of a simultaneous integrated boost (SIL) to the dominant intra-prostatic lesion (DIL) has also been shown to improve local control whilst minimising toxicity [11,12,13,14]

The Elekta Unity magnetic resonance linear accelerator (MRL) integrates the Philips 1.5 T magnetic resonance imaging (MRI) system (Best, The Netherlands) with the Elekta 7 MV linear accelerator (Stockholm, Sweden), enabling the delivery of MRI-guided adaptive radiotherapy (MRIgART) and the ability to account for interfraction motion of the prostate, seminal vesicles and associated organs at risk (OARs) throughout a course of prostate radiotherapy [15,16].

HERMES (NCT04595019) was the first completed study of two-fraction MRIgART with a gross tumour volume (GTV) boost in the treatment of men with intermediate to high-risk prostate cancer [17]. This groundbreaking trial combines the daily online MRIgART with clinical target volume (CTV) to planning target volume (PTV) margin reduction to safely deliver SBRT in two fractions with a simultaneous integrated boost (SIB) to the DIL.

The consequential effect of air in the rectum on rectal volume and prostate position is well documented [18], however the impact that air may have on dose distribution when treating a 1.5 T MRL has been less well explored. Electrons liberated through photon interactions with tissue will deflect in a magnetic field due to the Lorentz force [19]. At tissue–air interfaces this deflection causes electrons entering the air region to re-enter the tissue (electron return effect (ERE)) which causes dose to be deposited upstream of tissue air interfaces, potentially increasing the dose to the rectal wall in prostate cancer radiotherapy.

The ERE is therefore another factor which may impact the delivered dose during the online workflow. Despite daily enemas, MRIs taken during MRIgART to the prostate often demonstrates the presence of gas in the rectum, indicating that such an interface may exist, appear or even disappear during the online workflow, potentially altering rectal dose during prostate SBRT [20].

Here we investigate whether the estimated delivered dose matches the planned dose in the HERMES protocol, and whether the ERE has a clinically relevant impact on dose to the rectal wall.

## 2. Materials and Methods

### 2.1. Patient Population

In HERMES (NCT04595019), 46 men with intermediate or favourable high-risk prostate adenocarcinoma (MRI stage T2–T3a, Gleason 4+3 or less, maximum PSA 25 ng/mL) and a dominant lesion visible on multiparametric magnetic resonance imaging (mpMRI) were treated with SBRT on the MRL. Patients were randomised to either two fractions of SBRT with a boost to the visible tumour as defined on MRI or to five-fraction SBRT with no boost to the GTV. No rectal spacers were permitted in this trial. Participants were co-enrolled into the MOMENTUM observational registry [21]. The first five patients from each dose cohort were analysed in this study.

### 2.2. Pre-Treatment Workflow

Patients underwent a pre-treatment computed tomography (CT) and MRI to aid the generation of an offline reference plan, which is required for the online adaptive workflow. Patients were prescribed microlet enemas 2 days before and on the day of scanning and instructed to drink 350 mL of water just prior to scanning, all with the aim of achieving an empty rectum and comfortably full bladder. They were placed in a head-first and supine position with indexed knee, foot immobilisation and head support.

The contours and resultant reference plan were generated on a simulation MRI (MRI_sim_) acquired on the Unity MR-Linac (T2 3D Tra, 1 mm slice thickness), acquired with the same preparation and position as above. The prostate plus the proximal 1 cm of seminal vesicles (SV) was defined as CTV_psv_. Men classified as having upper intermediate (Gleason 4+3) and high-risk disease, were treated with a further CTV, defined as the prostate plus proximal 2 cm of SVs (CTV_sv_) [17]. A 3 mm margin was applied to the CTV(s) to create the PTV(s).

Participants in the 2-fraction group were prescribed 27 Gy to the GTV, the visible tumour on pre-biopsy mpMRI, with no margin applied. In the 2-fraction cohort, the PTV_psv_ and PTV_sv_ were prescribed 24 Gy and 20 Gy, respectively. In the 5-fraction cohort the PTV_psv_ and PTV_sv_ were prescribed 36.25 Gy and 30 Gy, respectively. There was no GTV boost in the 5-fraction group; however, the CTV_psv_ with no margin was treated to 40 Gy as per the PACE trial [22]. The target volumes are shown in Figure 1 and the prescribed doses displayed in Appendix A along with all of the dose constraints.

### 2.3. Treatment Workflow

All participants received 11-field intensity-modulated radiotherapy on the MR-Linac utilising the adapt-to-shape workflow, which has been described in detail previously [16]. All targets and organs at risk (OARs) were propagated from the reference plan to a daily session of MRI (T2 3D Tra) (MRI_session_) via deformable image registration, except GTV and urethra which were rigidly propagated as the GTV is less distinct on online imaging after the use of androgen-deprivation and the urethra is not likely to deform relative to the prostate on a day-to-day basis. The target and OARs were edited on the MRI_session_, and a new plan created. Online MR-based dose calculation was facilitated using a bulk density override method whereby patient-specific overrides were applied to the combined CTVs, bones, and the patient external contour as per a previously detailed methodology [23]. A second MRI (MRI_verif_) was acquired immediately prior to beam-on, and were there any visible displacements in the target anatomy these were corrected using an adapt-to-position workflow. A final MRI (MRI_post_) was taken once the treatment had been delivered, before acquiring this image patients were informed that the treatment had finished, and their comfort checked to ensure that they were happy to stay on the couch for the MRI_post_.

Each fraction in the 2-fraction regimen was delivered in two sequential sub-fractions to reduce the impact of intrafraction motion which might occur during a lengthy beam-on time [24]. Patients emptied their bladder after they had moved off of the couch before re-hydrating, if deemed appropriate when assessing the bladder filling rate on the MRI_sim_, and waiting 20 min to move back on to the couch for the second sub-fraction.

### 2.4. Dosimetric Analysis

The prostate, seminal vesicles, bladder and rectum were recontoured and the GTV and urethra structures repositioned offline on the MRI_verif_ and MRI_post_ for each fraction/sub-fraction in Monaco TPS (Elekta AB, Stockholm, Sweden, V5.40.01. Visible rectal gas was also contoured.

### 2.5. Dose Delivered

Dose calculations were performed retrospectively on the MRI_verif_ and MRI_post_ to calculate the estimated delivered dose in the 2-fraction and 5-fraction regimes and to establish the ERE-induced effect when considering the presence of air in and around the prostate/rectum interface, once using the standard bulk density override strategy, and then again with air included as an additional override structure.

Table 1 shows the target and OAR dose constraints which were analysed in each cohort.

The percentages of sub-fractions/fractions meeting the mandatory dose constraint for each specified target objective on the MRI_session_, MRI_verif_ and MRI_post_ were calculated.

The doses delivered to the targets and specified OARs on the MRI_session_, MRI_verif_ and MRI_post_ for each subfraction/fraction of every patient were measured [25]. In sub-fractions/fractions where the mandatory target dose constraint was not met the % volume of the target receiving the intended dose is reported.

The dose received by the OAR volumes for each patient was estimated by taking the mean of the daily dose achieved across the subfractions/fractions, in this way the dose is conservative in that it assumes the same part of the OAR is always receiving the maximum dose.

### 2.6. Electron Return Effect

The impact of air around the prostate/rectum interface on the rectal dose was evaluated by calculating the correlation coefficient between the volume of air present in the rectum and the difference in the resulting rectal D1 cm^3^, assuming the same volume of air was present though out all sub-fractions/fractions. The impact of the volume of air within 2 cm, 1 cm and 0.5 cm was measured and the associated correlation was measured on the MRI_verif_ and MRI_post_ of each sub-fraction/fraction.

The results were analysed using GraphPad Prism 10.1.2. The Shapiro–Wilk and Kolmogorov–Smirnov tests were carried out to test for normality. Normally distributed data are presented as a mean with 95% confidence intervals (CI). Skewed data are presented using the median with the range or interquartile range (IQR).

## 3. Results

Imaging was available for recontouring and plan adaption on the MRI_verif_ and MRI_post_ of every sub-fraction/fraction except for one 5-fraction treatment in which the MRI_post_ was not completed on one occasion due to the patient requesting to move off the couch after treatment deliver in order to pass urine.

### 3.1. Percentage of Target Volumes Meeting the Mandatory Dose Constraints

2-fraction and 5-fraction cohort

The percentage of sub-fractions/fractions which met the mandatory dose constraints for the specified GTV and CTV coverage in the 2-fraction and 5-fractions patients are shown in Table 2. Target coverage decreased from the MRI_session_ to MRI_verif_ and from the MRI_verif_ to the MRI_post_.

### 3.2. Target Coverage

2-fraction cohort

The D95% delivered to the targets for each sub-fraction of the 2-fraction cohort are shown for each patient in Figure 2. The doses received in each sub-fraction are shown in Appendix A.

In the sub-fractions where the mandatory target objective was not met, the median coverage was as follows: 86% (2 sub-fractions, range: 86–87%) for the GTV V27 Gy on the MRI_verif_, 87% (2 sub-fractions) for the GTV V27 on the MRI_post,_ 81% (2 sub-fractions, range: 76–87%) for the CTV V24 Gy on the MRI_post_ and 81% (1 sub-fraction) for the CTV V20 Gy on the MRI_post_.

5-fraction cohort

The D95% of the targets for each fraction of the 5-fraction cohort are shown for each patient in Figure 3. The dose received by each sub-fraction is shown in Appendix A.

The median target coverage, calculated from only those fractions where the mandatory constraint was not met, was as follows: 83% (3 fractions, range: 58–87%) for the CTV_psv_ V36.25 Gy on the MRI_post_, and 82% (2 fractions, range: 72–93%) for the CTV_sv_ V30 Gy on the MRI_post_.

### 3.3. OAR Received Dose

2-fraction cohort

The mean urethral dose met the mandatory urethral D10% constraint for each patient on the MRI_verif_ and MRI_post_. The mean bladder D5 cm^3^ and D15cm^3^ constraints were met in all cases except on the MRI_post_ for patients 3 and 4.

The accumulated rectal D1 cm^3^ < 20.8 Gy was exceeded on the MRI_verif_ and MRI_post_ in three of the patients. Figure 4 shows the mean rectal doses for each patient across all subfractions in the 2-fraction cohort. The data for each sub-fraction and mean dose to the OARs from all four fractions is shown Appendix A.

5-fraction cohort

In the 5-fraction cohort the rectal optimal constraint of D1 cm^3^ < 36 Gy was exceeded by patient 3 on the MRI_verif_ and MRI_post_ (Figure 5). The rectal mandatory constraint of D20% < 29 Gy was met for all patients at all time points. The results for each fraction and the mean dose to the OARs from all five fractions are shown in Appendix A.

### 3.4. Electron Return Effect

The difference in the rectal D1 cm^3^ (Gy) when calculated with and without the presence of air correlates with the volume of air in the rectum. The volume within 1 cm of the prostate volume showed the strongest correlation. The correlation coefficient (R) was strong (R = 0.69) and very strong (R = 0.90) in the 2-fraction (Figure 6) and 5-fraction cohorts, respectively (Figure 7).

The largest increase seen in a patient’s rectal D1 cm^3^ when calculated across all fractions/sub-fractions was 0.8 Gy in the 2-fraction cohort and 1.2 Gy in the 5-fraction cohort (Appendix A).

## 4. Discussion

In the first 10 patients treated within HERMES, all sub-fractions/fractions achieved the mandatory CTV target objectives on the MRI_verif_ and ≥88% of the sub-fractions/fractions achieved the mandatory target objective on the MRI_post_ when delivering 2- and 5-fraction SBRT with a 3 mm PTV margin and ATS workflow on the MRL. Reassuringly, coverage of the 2-fraction boost GTV, with no margin, was well achieved (89% of sub-fractions on the MRI_post_) when planned isotoxically to dose constraints, demonstrating that 27 Gy was delivered to the DIL in the majority of patients. The CTV likely acts as a margin by limiting dose fall off, which helps to improve GTV coverage in the absence of a PTV margin.

When delivering 2-fraction prostate SBRT with a boost, the bladder constraints were met in all but two patients on the MRI_post_. The rectal D1 cm^3^ proved the most exceeded constraint with the results being worse in the 2-fraction cohort than the 5-fraction cohort, however the constraints for the rectal D4 cm^2^ (2-fraction cohort) and D20% (5-fraction cohort) were more readily met.

There may be multiple mitigations for the findings. Firstly, patients were alerted when treatment delivery was finished and the MRI_post_ would be taken, therefore at this point there may have been large changes in the patient’s anatomy if the patient then relaxes or changes position on the treatment couch. These positional changes will not actually have occurred during treatment but may give the impression that less of a dose was delivered to the target and more of a dose was delivered to an OAR, than truly occurred.

Secondly, when considering the OAR dose to the rectum, the point dose measured is a worst-case scenario, assuming that the same 1 cm^3^ of rectum receives the high dose during each fraction. In truth, it is known that rectal motion means this may not be the case. Furthermore, in the 5-fraction cohort the optimal rather than mandatory constraint of 1 cm^3^ was calculated, in the 2-fraction cohort there was only a mandatory constraint to work to.

Despite accumulated OAR dose being exceeded in some patients, the HERMES interim analysis outcomes were promising; with only 10% of patients in the 2-fraction arm and 20% of patients in the 5-fraction arm experiencing acute ≥ 2 CTCAE GU toxicity [20]. There were no acute grade ≥ 3 GU toxicities or ≥2 GI toxicities. The interim analysis showed that the HERMES protocol can deliver 2- and 5-fraction prostate SBRT with levels of acute toxicity no worse than those seen in PACE B (CTCAE grade ≥ 2 GU toxicity of 15.3% and grade ≥ 2 GI toxicity of 29.2%) [22]. We await the results of the full acute toxicity analysis from HERMES.

The OAR constraints for 2-fraction group were created from previously published studies of 2-fraction HDR and, at the time of writing the protocol, the sole trial of 2-fraction external beam radiotherapy [26,27,28]. Whilst these studies suggest the HERMES dose constraints are safe, it is possible that they are in fact overly conservative. For example, if the linear–quadratic model holds down to two fractions, then the rectal constraint of V20.8 Gy < 1 cm^3^ equates to the equivalent dose in 2 Gy fractions (EQD2) of 55.74 Gy, with an α/β of 3 Gy, which is tighter than the constraints that are used for either 20 or 5 fractions.

It has been hypothesised that CTV-PTV margin reduction in the context of MRIgART further decreases the dose to the OARs and therefore may reduce the level of acute and late toxicity seen in CT-based radiotherapy. Margins needed for non-MRIgART have been suggested to be larger for patients with increased abdominal girth and image-guidance method is relevant to the risk of side effects [29,30,31]. MIRAGE (NCT04384770) investigated the benefits of margin reduction on toxicity for five fraction prostate SBRT [32]. Men received either CT-guided radiotherapy with a 4 mm CTV to PTV margin or MRIgART with a 2 mm CTV to PTV margin and gating software (ViewRay, Inc., Oakwood Village, OH, USA). There was a significant reduction in acute grade ≥ 2 genitourinary (GU) toxicity (24.4% vs. 43.4%, *p* = 0.01) and acute grade ≥2 GI toxicity (0% vs. 10.5%, *p* = 0.003) within the MRIgART arm. In the absence of gating, a 3 mm margin was adopted for this study with the addition of a manual hold being implemented if the prostate moved outside of the 3 mm PTV margin.

The introduction of comprehensive motion management (CMM) for the 1.5 T MRL will enable gating and baseline shift correction during beam-on time. Gating, as deployed in MIRAGE, initiates an automatic beam hold when the target moves outside of a pre-defined threshold. Baseline–shift–correction re-directs the beam to a target’s new position when motion occurs, allowing dose delivery to the target to continue. When implemented, CMM will allow the treating team to be more confident in achieving the target dose constraints from the MRI_session_ the time of the MRI_post_ and may facilitate additional margin reduction.

These results were calculated as they would be in the online setting; however, the impact of air on dose distribution was also explored, a factor not previously considered in the online workflow [25]. Our findings suggest that air within the rectum can increase the dose to the rectum when the worse-case effects of air are assumed. The dose increase may be clinically meaningful if dose is delivered in just 2 fractions, or the air is constantly present throughout a 5-fraction course. To help mitigate this potential uncertainty in dose delivery, our department has moved to a new beam arrangement designed to reduce the impact of the ERE.

There is ongoing debate regarding the cost of MRIgART and the length of time spent on the treatment couch [33]. This work adds to the expanding evidence supporting the move to increasingly hypofractionated regimes due to the socio-economic benefits to hospitals and patients, with a reduction in travel costs, re-location costs, time off work and improved toxicity outcomes for patients [34]. Furthermore, treatment on the MRL negates the need for fiducial markers and radiation exposure secondary to cone beam CT or X-rays, meaning that, although treatment requiring fiducials and/or CT guidance may provide a safe approach to implementing CTV-PTV margin reduction, it remains invasive and with the associated risks [35]. The MRL requires a multidisciplinary team for treatment and therefore treatment that is administer over just two days reduces the pressure of coordination of team members compared with 5 or 20 fraction treatments.

The limitations to this study are recognised by the authors. The size of the study is a potential limitation, with a total of 10 patients being contoured and analysed. However, 45 fractions were analysed, with analysis carried out on both the MRI_verif_ and MRI_post_ of each fraction (89 images in total) in order to give a good representation of the dosimetric implications of intrafraction motion.

The lack of control group is a limitation when trying to establish whether the MRL and the integration of adaptive radiotherapy is truly beneficial in delivering 2-fraction SBRT to the prostate. This could be addressed in the future using CT-based imaging, with or without CT-guided adaptation. The dose strategies for the two arms are different, with the 2-fraction cohort having a GTV-only boost, and the 5-fraction cohort receiving a boost of 40 Gy to the whole prostate (with no margin). The latter artificially elevates the dose, meaning the dose to the CTV is artificially elevated at baseline and can therefore fall further without violating the CTV dose objective. These factors limit the potential to compare the dose delivery between the two cohorts and compare it to a delivered dose on another platform.

This study examines intrafraction motion as measured on the MRI_verif_ and MRI_post_; however, prostate motion could be of a larger amplitude during treatment. Reassuringly, studies examining the motion of the prostate over time suggests that it follows a random walk model with the variance growing over time [36]. If this is indeed the case, then the MRI_post_ likely represents the worse possible scenario with the amplitude of motion being smaller during treatment. As previously discussed, it is possible that patient motion or relaxation after the beam is switched off may exaggerate the dosimetric effect of motion observed on the MRI_post_. As a safety solution, manual beam hold based on real-time cine MR images mitigates the possibility of the prostate moving outside of the 3 mm CTV-PTV margin during treatment.

The clinical outcomes supporting the safe reduction in CTV-PTV margins are not yet available. With margin reduction on the MRL being only recently introduced, there has not yet been an analysis of biochemical control. Indeed, with the biochemical control after SBRT in the PACE-B trial now exceeding 95%, a very large trial would be needed to demonstrate the non-inferiority of biochemical control for margin changes and is likely not feasible. With 68% of men in MIRAGE and all men in HERMES receiving hormonal treatment the data on PSA kinetics from these small trials are still maturing. A secondary end point of HERMES includes 5-year biochemical recurrence-free survival, which will help to confirm the safety of CTV-PTV margin reduction in the context of 2-fraction prostate MRIgART, along with other two fraction trials such as FORT (NCT04984343), iSMART (NCT05600400) and SABR-Dual (NCT0602789) [37,38]. Cost-effectiveness of 2-fraction vs. 5-fraction MRIgART, or longer schedules, is yet to be determined. The toxicity and oncological outcomes of these trials, and the HERMES trial, are currently immature but will be important in order to determine the future direction of prostate SBRT.

## 5. Conclusions

In HERMES the innovative delivery of 2-fraction MRIgART with a boost to the tumour is achievable and, for most patients, dose delivered closely matches dose planned despite intrafraction motion. The ERE was shown to have a potential impact on rectal dose suggesting that further studies are required to find out how to best mitigate these effects.

## Figures and Tables

**Figure 1 cancers-16-02073-f001:**
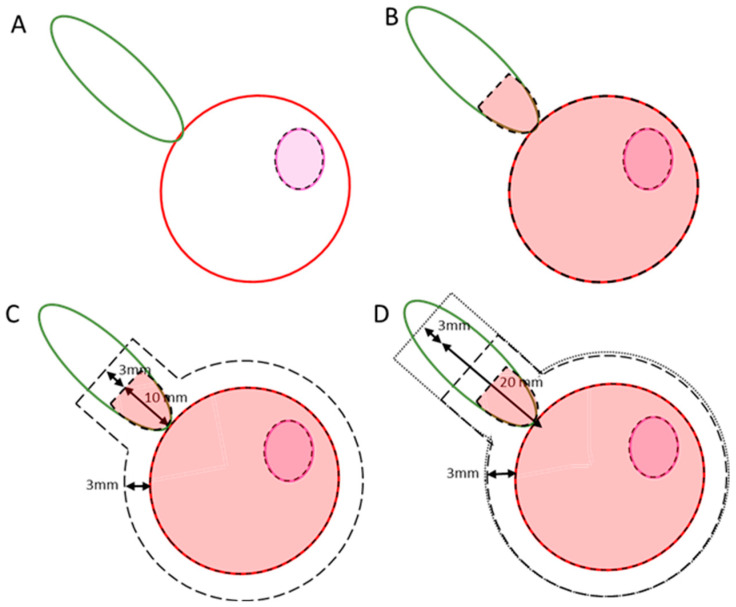
Targets with their associated margins. The prostate is shown with the red outline, and the seminal vesicles in green. The GTV is represented by the fuchsia outline. (**A**) shows how the GTV is treated without a PTV margin, receiving 27 Gy in the 2-fraction group, shown with the pink shadowing. In (**B**), the black dashed line represents the CTVpsv with the red shadowing representing the boosting to 40 Gy in the 5-fraction group. (**C**) shows PTVpsv, with the outer black dashed line and no fill; this receives 24 Gy in the 2-fraction arm and 36.25 Gy in the 5-fraction arm. In (**D**) the dotted line, with no fill, represents the PTVsv; this receives 20 Gy in the 2-fraction group and 30 Gy in the 5-fraction group.

**Figure 2 cancers-16-02073-f002:**
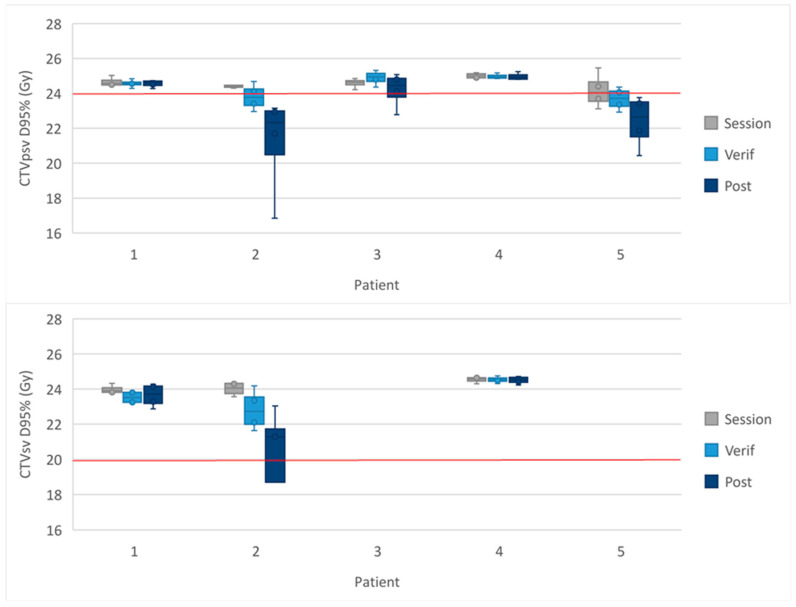
Dose to the CTVpsv D95% and CTVsv D95% in the 2-fraction cohort. The spread of the D95% received by the targets for each sub-fraction for every patient in the 2-fraction cohort. The whiskers show the range, the box the IQR and the midline the median. The red line represents the protocol-defined constraint.

**Figure 3 cancers-16-02073-f003:**
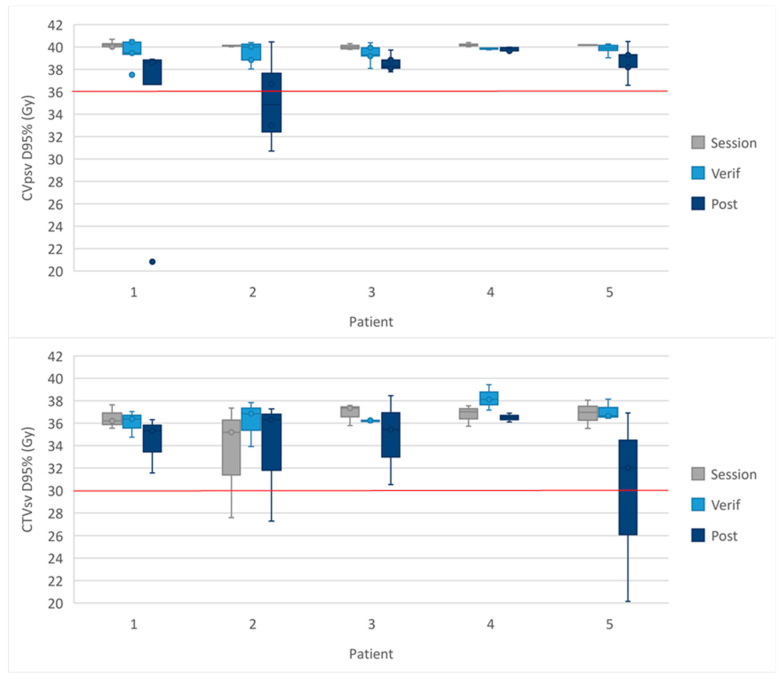
Dose to the CTVpsv D95% and CTVsv D95% in the 5-fraction cohort. The spread of the D95% received by the targets for each patient in the 5-fraction cohort. The whiskers show the range, the box the IQR and the midline the median. The red line represents the protocol-defined constraint. The CTV_psv_ in the MRI_session_ often starts at 40 Gy due to the additional CTV_psv_ V40 Gy > 95%.

**Figure 4 cancers-16-02073-f004:**
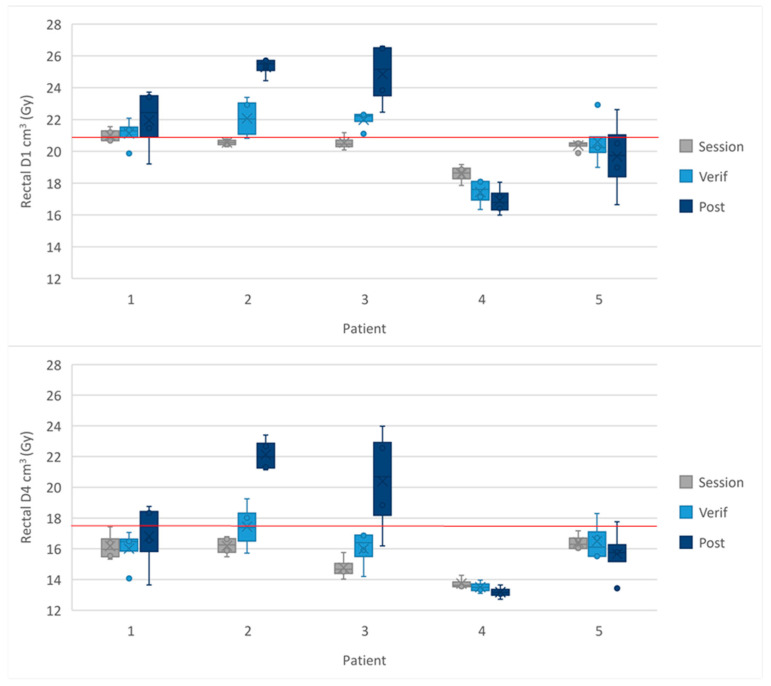
Dose to the rectal D1 cm^3^ and D4 cm^3^ in the 2-fraction cohort. The cross in the box plot represents the accumulated mean dose calculated from all 4 sub-fractions for each patient. The whiskers show the range and the box shows the IQR. The red line represents the protocol-defined constraint.

**Figure 5 cancers-16-02073-f005:**
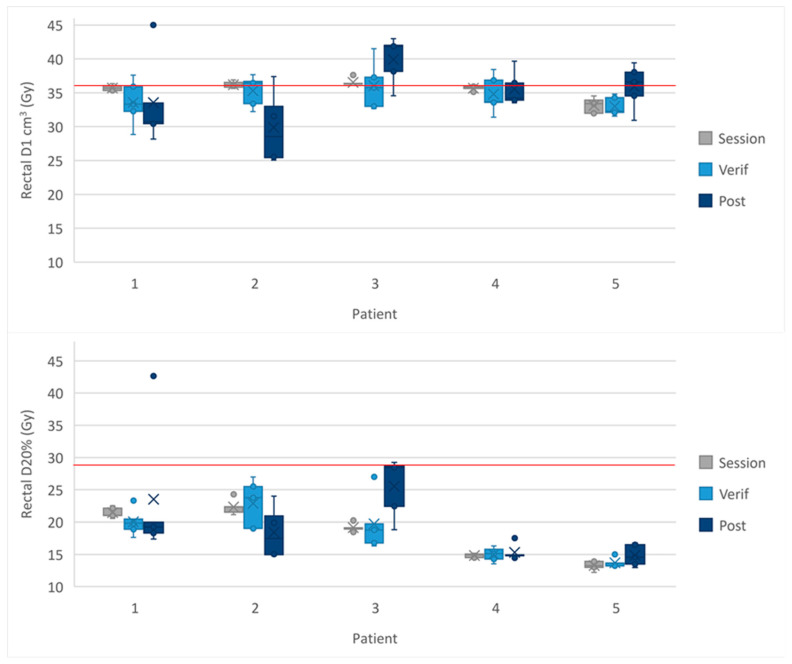
Dose to the rectal D1 cm^3^ and D20% in the 5-fraction cohort. The cross in the box plot represents the accumulated mean dose calculated from all 5 fractions for each patient. The whiskers show the range and the box shows the IQR. The red line represents the protocol-defined constraint.

**Figure 6 cancers-16-02073-f006:**
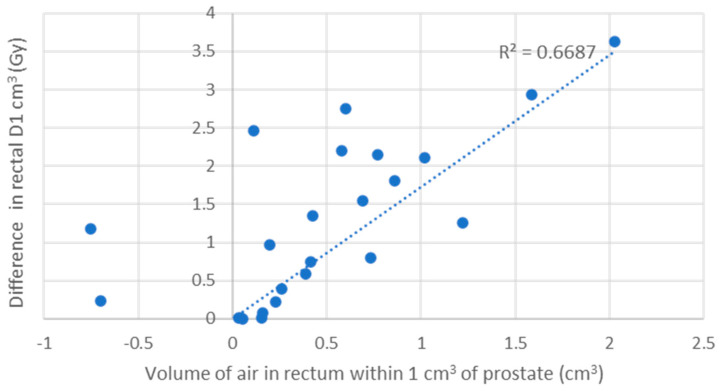
Correlation between volume of air in the rectum and change in dose to rectal D1 cm^3^ in the 2-fraction cohort.

**Figure 7 cancers-16-02073-f007:**
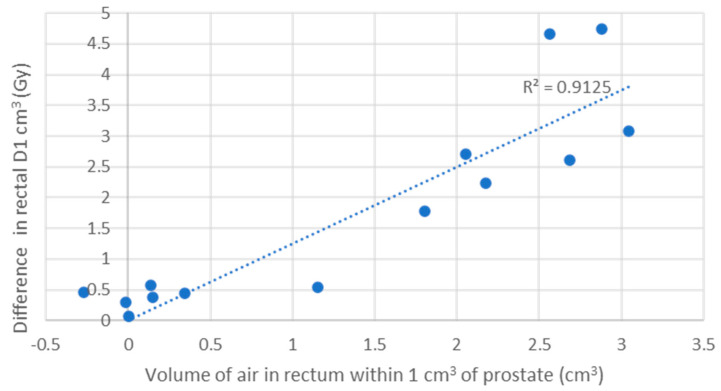
Correlation between the volume of air in the rectum and the change in dose to rectal D1 cm^3^ in the 5-fraction cohort. The linear correlation coefficient between the difference in the rectal D1 cm^3^ (Gy) calculated with and without the presence of air and the volume of air in the rectum within 1 cm of the prostate volume in the 2-fraction cohort. If the volume of air in the rectum remained present throughout each sub-fraction it is estimated that for each 1 cm^2^ of air the dose to the rectal wall would be expected to increase by 1.25 Gy over the full course of treatment. R = 0.90. Only fractions with air within 1 cm of the PTV were included.

**Table 1 cancers-16-02073-t001:** Target dosimetric criterion for each of the doses analysed for the targets (A) and OARs (B) in the 5-fraction and 2-fraction cohorts.

A.	Target Dosimetric Criterion	
Structure	2-fraction	5-fraction
GTV_27	V27 Gy > 95% (−5%)	NA
CTV_psv_	V24 Gy > 95% (−5%)	V36.25 Gy > 95%
CTV_sv_ (lower high-risk patients)	V20 Gy > 95% (−5%)	V30 Gy > 95%
B.	OAR Dosimetric Criterion	
Structure	2-fraction	5-fraction
Urethra	D10% < 26 Gy (+1 Gy)	D50% < 42 Gy
Bladder	D5 cm^3^ < V20.8 Gy	D5 cm^3^ (+5 cm^3^) < V37 Gy
	D15 cm3 < V14.6 Gy	D40% < 18.1 Gy
Rectum	D1 cm^3^ < 20.8 Gy	D1 cm^3^ (+1 cm^3^) < 36 Gy
	D4 cm^3^ < 17.6 Gy	D20% < 29 Gy

The dose received by the CTV was measured rather than that delivered to the PTV. Figures in parentheses represent acceptable deviations (mandatory constraints).

**Table 2 cancers-16-02073-t002:** The percentage of target volumes meeting the mandatory dose constraint across all sub-fractions/fractions.

			Mandatory Constraints Met (%)		
Cohort	Target	Dosimetric Criterion	Session	Verif	Post
2-fraction	GTV	V27 Gy > 95% (−5%)	100%	89%	89%
	CTV_psv_	V24 Gy > 95% (−5%)	100%	100%	90%
	CTV_sv_	V20 Gy > 95% (−5%)	100%	100%	92%
5-fraction	CTV_psv_	V36.25 Gy > 95%	100%	100%	88%
	CTV_sv_	V30 Gy > 95%	100%	100%	92%

The dosimetric criteria shown in parentheses represent the mandatory constraint as opposed to the optimal constraint of 95%.

## Data Availability

Formal requests for data sharing are considered in line with ICR-CTSU procedures, with due regard given to funder and sponsor guidelines. Requests are undertaken via a standard proforma describing the nature of the proposed research and extent of data requirements. Data recipients are required to enter a formal data sharing agreement that describes the conditions for release and requirements for data transfer, storage, archiving, publication, and intellectual property. Requests are reviewed by the Trial Management Group (TMG) in terms of scientific merit and ethical considerations including patient consent. Data sharing is undertaken if proposed projects have a sound scientific or patient benefit rationale, as agreed by the TMG and approved by the Independent Data Monitoring and Steering Committee as required. Restrictions relating to patient confidentiality and consent will be limited by aggregating and anonymising identifiable patient data. Additionally, all indirect identifiers that could lead to deductive disclosures will be removed in line with Cancer Research UK Data Sharing Guidelines.

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
