# Peer review of "Stereotactic Body Radiotherapy (SBRT) to Localised Prostate Cancer in the Era of MRI-Guided Adaptive Radiotherapy: Doses Delivered in the HERMES Trial Comparing Two- and Five-Fraction Treatments"

_cancers, 2024, doi:10.3390/cancers16112073_

Round 1

Reviewer 1 Report

Comments and Suggestions for Authors

The manuscript is interesting but some points could be improved

1. The Authors should consider if the use of spacer between rectum and prostate cloud reduce side effectcs of radiotheraphy (Pepe P, Tamburo M, Pennisi M, Marletta D, Marletta F. Clinical Outcomes of Hydrogel Spacer Injection Space OAR in Men Submitted to Hypofractionated Radiotherapy for Prostate Cancer. In Vivo. 2021 Nov-Dec;35(6):3385-3389. doi: 10.21873/invivo.12637. PMID: 34697173; PMCID: PMC8627770)

2. The Authors should underline in the conclusions the oncological and functional outcomes needs a longer follow up

3. The Authors should discuss the advantages in terms of cost-effectiveness of two radiotherapy sessions

Author Response

Thank you for your helpful comments. 

1) We agree that rectal spacers should be mentioned. We have added that rectal spacers were not permitted in this trial and in the subsequent toxicity paper (currently in preparation) will reflect on whether this could have altered toxicity. 

2) Thank you. We have added the following sentence to the Discussion "The toxicity and oncological outcomes of these trials, and the HERMES trial, are as yet immature but will be important to determine the future direction of prostate SBRT."

3) We have added the following sentence to the discussion "Cost-effectiveness of 2-fraction vs 5-fraction MRIgART, or longer schedules is yet to be determined"

Reviewer 2 Report

Comments and Suggestions for Authors

Dear authors,

I read with interest your paper. The topic is complex and you conceived well the study design. I have a few comments before publication:

1) In lines 68-71 you mentioned the Lorentz force as responsible for a higher dose deposit in the rectal wall. Can you explain better what is that? You refer to charged particles. Since you use photon beam-based radiotherapy, I think you're referring to the electrons generated by the photoelectric and Compton effects. Did I understand right? However, this issue must be clarified for readers.

2) In lines 122-124 I don't understand why you didn't propagate the GTV and urethra too. You should explain this, here or in the discussion.

3) In lines 160-161 the median dose you referred to contrasts with the D90%/D95% definition, which are also doses and not volumes as I would have expected to read here.

4) In line 179: "...except for one 5-fraction treatment when the MRIpost was not completed due to the length of time the patient requesting to get off the couch after treatment deliver to pass urine". Did you collect no MRIpost for the whole course of RT for that patient or just one MRIpost of the five planned? Please, clarify this.

5) The main aim of your dosimetric study also deals with the inherent risk of treatment-related adverse events, especially radiation proctitis, which strangely is not discussed throughout the paper. You could cite PMID: 33813420 and PMID: 33910815 as pertinent literature on the concerns of this risk and the current gain from the use of IGRT and other technical tricks, till high-precision radiation delivery techniques like yours.

6) I was fascinated by your insight into the PTV margin issue. I invite you to discuss the related findings in PMID: 37648318.

7) I'm not sure of the detrimental effect of any rectal gas. As discussed in the discussion of PMID: 33813420, the effect of the presence of gas within the rectum could be minimal and even protective on the rectal wall. Accordingly, as discussed in PMID: 33910815, the rectal balloon is one of the technical tricks currently used to reduce the rectum dose. These two opposite theories, given the lack of in vivo dosimetry, should be thoroughly addressed to ultimately define which of the two is true.

Comments on the Quality of English Language

Minor typos throughout the paper.

Author Response

Many thanks to the Reviewer for the time taken to thoroughly review our paper. It is much appreciated. Please find our responses and corrections below:

1) We have added some further detail on Lorentz force and expanded this paragraph below:

"Electrons liberated through photon interactions with tissue will deflect in a magnetic field due to the Lorentz force. At tissue-air interfaces this deflection causes electrons entering the air region to re-enter the tissue (electron return effect (ERE) which causes dose to be deposited upstream of tissue air interfaces, potentially increasing the dose to the rectal wall in prostate cancer radiotherapy."

2) Thank you we have clarified below:

"All targets and organs at risk (OARs) were propagated from the reference plan to a daily session MRI (T2 3D Tra) (MRIsession) via deformable image registration, except GTV and urethra which were rigidly propagated as the GTV is less distinct on online imaging, after the use of androgen-deprivation and the urethra is not likely to deform relative to the prostate day-to-day."

3) Thank  you for picking up this important point. We have clarified in the methods as follows:

"In sub-fractions/fractions where the mandatory target dose constraint was not met the % volume of the target receiving the intended dose is reported."

4)We agree this was ambiguous. We have clarified that this issue only affected one fraction for this one patient. 

5) We agree that the risk of toxicity is important, but decided not to include a discussion of this, as the focus of this paper is  about dosimetry, with no new reporting of the toxicity outcomes of the trial. An entire paper could be given over to discussing predictors and protectors for rectal toxcity. In fact,the clinical outcomes of the HERMES trial are currently being analysed and we will definitely include a full discussion of factors determining toxicity after 5- and 2-fraction SBRT in that paper. 

6) Thank you, we have added the suggested reference 

7)You are correct that the presence of rectal gas, a rectal obturator or a rectal balloon could push the posterior wall of the rectum further away from the dose. However our previous work with the PROSPARE device has suggested that the obturator overall worsens rectal dose by pushing the anterior wall more reliably into the high dose. For this paper, however the focus is on the effect of the Lorentz force rather than any geometric change to the rectum, per se. We wholeheartedly agree with the reviewer that further study on this matter is warranted. 

We apologise for the typos and hope we have caught them all now. We have reformatted and revised the references, partly in response to editorial comments about self-citation. 

With many thanks

Round 2

Reviewer 2 Report

Comments and Suggestions for Authors

Thank you for having clarified the issues underscored by me. In my opinion, your paper is now suitable for publication.